# Health Professionals' knowledge and practice on basic life support and its predicting factors in Ethiopia: Systematic review and meta-analysis

**Worku Necho Asferie**[1]*, **Demewoz Kefale**[1], **Amare Kassaw**[1], **Amare Simegn Ayele**[2], **Gedefaye Nibret**[2], **Yohannes Tesfahun**[3], **Habitamu Shimels Hailemeskel**[1], **Solomon Demis**[1], **Shegaw Zeleke**[4], **Tigabu Munye Aytenew**[4]

1 Department of pediatric and neonatal Nursing, College of Health Science, Debre Tabor University, Debre Tabor, Ethiopia, 2 Department of Midwifery, College of Health Science, Debre Tabor University, Debre Tabor, Ethiopia, 3 Department of Emergency and Critical Care Nursing, College of Health Science, Debre Tabor University, Debre Tabor, Ethiopia, 4 Department of Nursing, College of Health Science, Debre Tabor University, Debre Tabor, Ethiopia

* workunecho@gmail.com

**Data Availability Statement:** All necessary data and supplementary materials were included in the manuscript

## Abstract

### Background

Basic Life Support (BLS) is a sequence of care provided to patients who are experiencing respiratory arrest, cardiac arrest, or airway obstruction. Its main purpose is to maintain the airway, breathing, and circulation through CPR. This review aimed to estimate the pooled prevalence of Health Professionals' knowledge and practice on basic life support in Ethiopia.

### Method

Eligible primary studies were accessed from international database (PubMed, Google Scholar, Hinari databases) and grey literatures found in online repositories. The required data were extracted from those studies and exported to Stata 17 for analysis. A weighted inverse-variance random-effects model and Der Simonian-Laird estimation method were used to compute the overall pooled prevalence of Health Professional's knowledge, practice of basic life support and its predictors. Variations across the included studies were checked using forest plot, funnel plot, $I^2$ statistics, and Egger's test.

### Result

A total of 5,258 Health Professionals were included from 11 studies. The pooled prevalence of knowledge and practice outcomes on basic life support in Ethiopia were 47.6 (95% CI: 29.899, 65.300, $I^2$: 99.21%) and 44.42 (95% CI: 16.42, 72.41, $I^2$: 99.69) respectively. Educational status of the Professional's was significantly associated with knowledge outcome. Those who had degree and above were 1.9 times (AOR: 1.90 (1.24, 2.56)) more likely knowledgeable on basic life support than under degree.

**Funding:** The author(s) received no specific funding for this work.

**Competing interests:** The authors have declared that no competing interests exist.

**Abbreviations:** AEP, automated external defibrillators; AOR, Adjusted odds ratio; BLS, Basic life support; CPR, Cardiopulmonary Resuscitation; JBI, Joanna Briggs Institute; IHCA, In-hospital cardiac arrest; ICU, Intensive care unit; PRISMA, Preferred reporting for items in systematic review and meta-analysis..

## Conclusion

The overall pooled estimates of Health Professionals knowledge and practice on basic life support was considerably low. The educational status of the Health Professionals was significantly associated with knowledge outcome. The Health Professionals and responsible stakeholders should focus on the basic life support at Health Institutions. The professionals should advance their knowledge and skill on basic life support for the patients.

## Background

Basic Life Support (BLS) is a sequence of care provided to patients who experiencing respiratory arrest, cardiac arrest, or airway obstruction. Its main purpose is to maintain the airway, breathing, and circulation through cardiopulmonary resuscitation (CPR) [1]. It requires knowledge and skills in CPR, using automated external defibrillators (AED) and relieving airway obstructions in patients of every age [2–4].

The BLS program is designed to deliver knowledge to a wide range of healthcare professionals about several life-threatening emergencies. It also demonstrates training to provide CPR, use an AED, and relieve choking in a safe, timely, and effective manner to treat those emergencies. It is mandatory for all healthcare professionals to have comprehensive knowledge and skill about BLS [5, 6].

Cardiopulmonary resuscitation is a series of emergency lifesaving actions which is performed in an effort to manually resuscitate a person in cardiac arrest. It is a critical component of basic life support (BLS) as the first-line response to cardiac arrest before defibrillation and advanced life support become available [7]. It helps to preserve intact brain function until further measurements are taken to restore spontaneous blood circulation and breathing in a person who is in cardiac arrest. It is recommended in those who are unresponsive with no breathing or abnormal breathing [8].

In-hospital cardiac arrest (IHCA) is a major adverse event for hospitalized patients with a reported incidence of 1.6/1000 admissions in European countries [9]. Early recognizing and intervention saves lives of the patients [10]. Adequate knowledge and skills of health care providers with regard to the manoeuvre and techniques of CPR to prevent irreversible organ damages and improves the chances of survival of cardiac arrest victims [11, 12]. But, Some studies done in Africa showed most of the Health care providers had poor knowledge and practice regarding to basic life support/resuscitation at Hospitals [13–16].

Different factors affect the knowledge and skills of Health Professionals while conducting basic life support for the patients admitted at ICU like training of the resuscitation, years of clinical experience, educational status of the professionals and frequently involving in the resuscitation activities [16–18]. There was no study conducted the pooled estimates of the related topics in Ethiopia. Therefore, this study aimed to identify the pooled estimates of Health Professionals' knowledge and practice on basic life support and associated factors in Ethiopia.

## Methods

This systematic review and meta-analysis was carried out using a methodology of Preferred Reporting Items for Systematic Review and Meta-analysis (PRISMA) [19]. It was carried out by conducting a systematic synthesis of the pertinent primary studies on the Knowledge and

practice of Health Professionals on basic life support in Ethiopia. The review protocol has been submitted for registration in an international prospective register for systematic reviews.

## Searching strategy

For explicit presentation of the reviewing questions and searching criteria we followed adapted PICO or "PEO" that (Population, Exposure and Outcomes) for creating the MeSH Terms to retrieve the potential studies in the database inclusion. Based on this;

a. **Population**: Health Care Providers, Health Professionals.

b. **Exposure:** basic life support/resuscitation

c. **Outcome**: Knowledge and Practice of Health Professionals' on basic life support

**Study Design:** Observational studies and
**Setting**: Ethiopia
**Our Research question** "What is the national prevalence of Health Professional's knowledge and practice on basic life support in Ethiopia?"

Two approaches were followed to search potentially relevant studies. The electronic database search (PubMed, Google Scholar, Hinari, and Institution research repositories) and the manually search of the reference list of the previous prevalence studies were carried out to retrieve more articles. "Health Professionals", "Health Care Providers", "Knowledge", "Skill", "Practice", "Resuscitation", "Cardiopulmonary Resuscitation", "Basic life Support" and "Ethiopia" were the key terms mostly used for retrieving reputable articles from database both using separation and in-combination with the balloon operators like "OR" or "AND", Truncations (. . ..*) and Phrase (. . ..). The articles were searched from 01/01/2023 to 8/20/2023. Finally, all studies which were in agreement with the review question were retrieved and screened for inclusion criteria

## Eligibility

**Inclusion and exclusion criteria.** Primary studies of any design that described the prevalence, proportion, and extent of knowledge and practice of basic life support among Health Professionals were included in this analysis. But, primary studies were excluded for any of the following reasons: (a) there was no information on the prevalence of knowledge and practice outcome; (b) there was no full text; (c) there was a low quality score; (d) the full text of the article was not accessible after 3 emails were sent to the corresponding author; and (e) there were no narrative reviews, editorials, correspondence, abstracts, or methodological studies.

All retrieved studies were independently evaluated for eligibility by two authors (W. A and T. A), and any disagreements or inconsistencies were settled by the involvement of a third author (A.K), who broke the disagreement.

## Outcome variable measurements

The knowledge and practice of basic life support (defined here as knowledge of resuscitation, knowledge of cardiopulmonary resuscitation, knowledge of basic life support and practice of resuscitation, cardiopulmonary resuscitation and basic life support) among Health professionals was the main outcome variable. Those Health care professionals who score 70% and above for the knowledge measuring items were considered as Good knowledge otherwise poor knowledge [20, 21]. Those Health care providers who score 60% of the practice measuring items of basic life support would considered having adequate practice [22]. The second outcome was to assess factors that affect the pooled knowledge and practice on basic life support.

The association of outcome and factors was examined using odd ration that was calculated by two by table (or = ad/bc).

## Study screening and selection

Search results were first downloaded into Endnote version 7 and duplicates were removed. Then, selection of studies was conducted in 2 stages. First, title and abstract screening was done. Then, full-text reviewing was conducted. Through title and abstract screening by 2 independent authors (WA and TA), studies that mentioned the prevalence /magnitude/proportion knowledge and practice on basic life support among Health Professionals were selected for full text review. Then, from full-text reviewing, any article classified as potentially eligible by either author was considered as a full text and screened by both authors independently. At times of disagreement where a consensus could not be reached between the authors, a third author (AK) reviewed and resolved the disagreements

## Critical appraisal and reliability checkup

After screening the relevant studies, selected studies were critically appraised for methodological validity using Joanna Briggs Institute (JBI) appraisal tool for prevalence studies [23]. The tool had a total of 9 questions (Q1-Q9) and those studies with positive answer of more than 50% of the tool (i.e. 'Yes' for 5 or more question of JBI tool) were included for this meta-analysis (Table 1).The risk of bias for each primary studies would evaluate using the adopted tool from hoy et al. [24] (Table 2). The scoring was done by 2 investigators (SD and DK) with the discrepancies were resolved with discussion and consensus. When the disagreement between the 2 authors was not resolved with discussion, the third author (AA) was involved as a breaker. During critical appraisal of each primary study, more emphasis was given to the appropriateness of the study objectives, study design, sampling technique, data collection technique, statistical analysis, any sources of bias and its management technique.

## Data extraction

The investigators extracted the required data using a pre-tested data extraction format using Microsoft excel 2010. The following information was taken from the studies: the first author, the region where the study was conducted, specific study area, study design, study publication year, study sample size, response rate of the study, and the prevalence of knowledge and practice of basic life support. Additionally, variables which were significantly associated with each primary studies extracted considering the following points: adjusted odd ration and their confidence interval to computed their Standard Errors and log odd ration that used for the final analysis to pooled signicant variables for the reviewed question. Any discrepancies between the two authors regarding the data extraction process were resolved through discussion and agreement. Involving a third reviewer also helped to address the variation.

## Stastical analysis

The necessary data were extracted using Microsoft excel 2010 and exported to Stata 17 (STATA Corporation, College Station Texas) for analysis. The primary study was summarized using table and forest plot. The Authors calculated the standard errors of the prevalence of knowledge and practice for each original articles using binomial formula. We check the level of Heterogeneity among the reported prevalence of the studies using the Cochrane $Q^2$ and $I^2$ stastics [25, 26]. The heterogeneity was quantified high (considerable), moderate, low as 75% and more, 50–75% and 25% and less respectively. The random effects model was used to

**Table 1. Quality assessment of the included studies using the Joanna Briggs Institute (JBI) quality appraisal criteria, Ethiopia, 2023.**

**Quality Appraisal for included studies in this systematic review and meta-analysis in Ethiopia, 2023**

| S/N | Author (Year) | Criteria | | | | | | | | Scores | Overall quality |
|---|---|---|---|---|---|---|---|---|---|---|---|
| | | Clearly defined inclusion criteria | Describing the study settings participants | Valid &reliable exposure measurement | Objective &standard criteria for measurement | Identified confounder | Strategies to deal with confounder | Valid & reliable outcome measurement | Appropriate statistical analysis | | |
| 1. | Kelkay et al. 2018 [35] | N | Y | Y | Y | N | N | Y | Y | 5 | Low risk |
| 2. | Mersha. A et al.,2020 [39] | Y | Y | Y | Y | N | Y | Y | Y | 7 | Low risk |
| 3. | Mersha AT et al., 2020 [18] | N | Y | Y | Y | N | Y | Y | Y | 6 | Low risk |
| 4. | (Abebe et al.2021) [37] | Y | Y | Y | Y | N | Y | Y | Y | 7 | Low risk |
| 5. | Bizuwork K et al, 2019 [36] | N | Y | Y | Y | N | Y | Y | Y | 6 | Low risk |
| 6. | Bikamo .E, 2021 [40] | Y | Y | Y | Y | N | Y | Y | Y | 7 | Low risk |
| 7. | Benwu.MK, 2021[42] | N | Y | Y | Y | N | Y | Y | Y | 6 | Low risk |
| 8. | Abebaw M et al. . . 2022 [41] | Y | Y | Y | Y | N | Y | Y | Y | 7 | Low risk |
| 9. | Bekele et al, 2021[38] | Y | Y | Y | Y | N | Y | Y | Y | 7 | Low risk |
| 10. | Sintayehu et al,2020 [43] | Y | Y | Y | Y | N | Y | Y | Y | 7 | Low risk |
| 11. | Getuta, 2022 [17] | Y | Y | Y | Y | N | Y | Y | Y | 7 | Low risk |

Note: Y- Yes, N- No

estimate the der Simonian and Laird's pooled effects since test statics showed there was a considerable heterogeneity among studies ($I^2$ = 99.21, P = 0.000 for knowledge and $I^2$ = 99.69, P = 0.0019 for practice). The publication bias was conducted using subjectively by funnel plot and objectively using egger's test with 5% significant level. In egger's test p-value less 0.05 indicates the presence of publication bias while greater than 0.05 indicates the absence of bias [27, 28]. If publication bias is noticed in the Random effects model, the estimate is determined by using Duval and Tweedie's trim and Fill analysis. In addition, subgroup analysis was done using region of studies in order to reduce the random heterogeneity between the estimates of the primary studies.

# Result

## Literature searching findings

A total of 917 articles were retrieved after a thorough search of both published and unpublished sources. Out of 917 articles, 913 were collected from databases. The remaining four

**Table 2. Risk of bias assessment of the included studies.**

| S/N | Author (Year) | Criteria | | | | | | | | | | Scores | Overall risk of bias |
|---|---|---|---|---|---|---|---|---|---|---|---|---|---|
| | | External validity | | | | Internal validity | | | | | | | |
| | | Q1 | Q2 | Q3 | Q4 | Q5 | Q6 | Q7 | Q8 | Q9 | Q10 | | |
| 1. | Kelkay et al. 2018 | N | Y | Y | Y | Y | N | Y | Y | Y | Y | 8 | Low risk |
| 2. | Mersha. A et al.,2020 | N | Y | Y | Y | Y | Y | Y | Y | N | Y | 8 | Low risk |
| 3. | Mersha AT et al., 2020 | N | Y | Y | Y | Y | Y | Y | Y | N | Y | 8 | Low risk |
| 4. | (Abebe et al.2021) | N | Y | N | Y | Y | Y | Y | Y | Y | Y | 8 | Low risk |
| 5. | Bizuwork K et al, 2019 | N | Y | Y | Y | Y | Y | Y | Y | N | Y | 8 | Low risk |
| 6. | Bikamo .E, 2021 | N | Y | Y | Y | N | Y | Y | Y | Y | Y | 8 | Low risk |
| 7. | Benwu.MK, 2021 | N | Y | Y | Y | Y | N | Y | Y | Y | Y | 8 | Low risk |
| 8. | Abebaw M et al, 2022 | N | Y | Y | Y | Y | Y | Y | Y | N | Y | 8 | Low risk |
| 9. | Bekele et al, 2021 | N | Y | Y | Y | N | Y | Y | Y | Y | Y | 8 | Low risk |
| 10. | Sintayehu et al,2020 | N | Y | Y | Y | Y | Y | Y | Y | N | Y | 8 | Low risk |
| 11. | Getuta, 2022 | N | Y | Y | Y | Y | Y | Y | Y | Y | N | 8 | Low risk |

**Note:** Y: yes, N: No

articles were obtained from institutional research repository at Addis Abeba University. Of the 913 articles found through database searching, 223 were found through Google Scholar, 325 through PubMed, and 365 through Hinari. About 857 articles were eliminated due to duplication, different countries and different interest question. Additionally, 6 articles [29–34] were excluded due to target group and methodological differences. Finally, 11 articles were included for this systematic review and meta-analysis (**Fig 1**).

## Characteristics of the original studies

The characteristics of 11 studies included in this review have been described in details in Table 3. All studies were used cross sectional study design with sample size ranges of 143, in Amhara regional state and 445 in SNNP. These studies were conducted from 2014 to 2022. In this meta-analysis, a total 3849 sample size was used to estimate the overall prevalence of Health professional's Knowledge and practice of resuscitation in Ethiopia. The 11 studies were conducted at different regions of Ethiopia; Amhara region 5 studies [18, 35–38], 1 South Nation Nationality and People (SNNP) [39], 2 in Addis Ababa [40, 41], 1 in Harar [14], and 1 in Tigray [42].

## Risk of bias assessment

The tool developed by Hoy *et al* was used to assess the risk of bias for each included study [24]. The tool consists of ten items that assess four areas of bias; internal validity and external validity. Items 1–4 evaluate selection bias, non-response bias and external validity. Items 5–10 assess measure bias, analysis-related bias and internal validity. Accordingly, of the total of the twenty-six included studies, twenty-two studies scored eight of ten questions and the four studies also scored seven of ten questions. Studies were classified as ″low risk″ if eight and above of ten questions received a ″Yes″, as ″moderate risk″ if six to seven of ten questions received a ″Yes″ and as ″high risk″ if five or lower of ten questions received a ″Yes″. Therefore, all included studies had low risk of bias (**Table 2**).

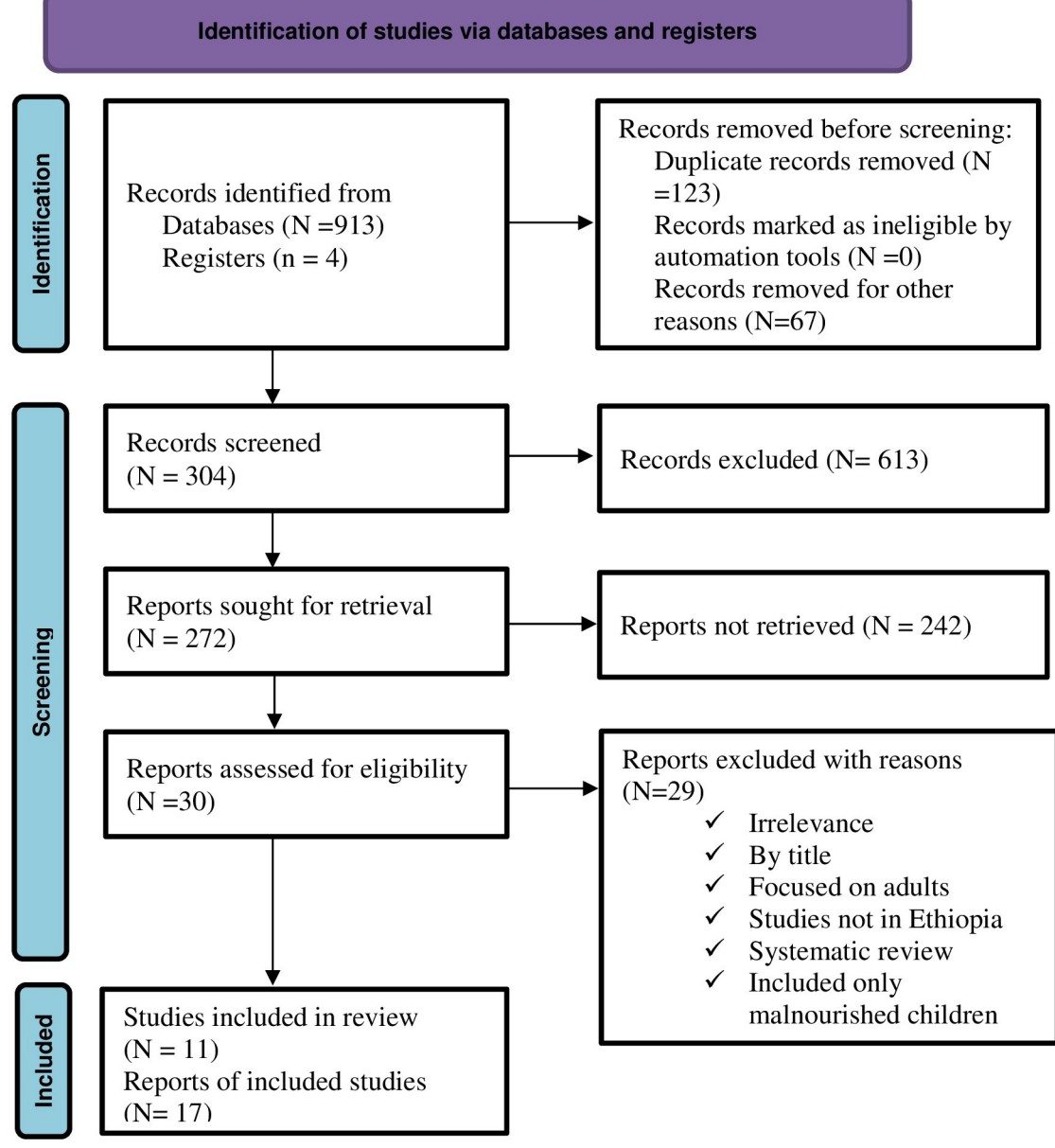

**Fig 1. PRISMA flow diagram of article selection for systematic review and meta-analysis of the Health Professionals' knowledge and practice on basic life support and its predicting factors in Ethiopia.**

## Meta-analysis

### Pooled prevalence of Health Professionals' knowledge on basic life support/ resuscitation

Before meta-analyzing the effect sizes of included studies, the presence of statistical variability between the included studies was checked using both visual inspection of forest plot and statistical test of variation. There was high/considerable heterogeneity among the included studies in pooled prevalence of Health Professionals' knowledge regarding to resuscitation. The Stata generated statistical test of variation (I squared statistics = 99.21% and Chi-squared = 1015.18 (d.

**Table 3. Descriptive summary of 11 studies reporting the knowledge, practice and associated factors of resuscitation/basic life support among Health Professionals in Ethiopia included in the systematic review and meta-analysis, 2023.**

| Author (year) | region | Publication Year | Sample size | response rate % | Quality score | Prevalence (95% CI) | Data Collection Techniques | Fund |
|---|---|---|---|---|---|---|---|---|
| Kelkay et al. 2018 [35] | Amhara | 2018 | 397 | 97.7 | 5 | 38.60 (33.81, 43.39) | Self-administered | UOG |
| Mersha. A et al.,2020 [39] | SNNPR | 2020 | 445 | 96.4 | 7 | 76.2 (72.24, 80.16) | Self-administered | AU |
| Mersha AT et al., 2020 [18] | Amhara | 2020 | 424 | 95.7 | 6 | 25.1(20.97, 29.23) | Self-administered | Not reported |
| (Abebe et al.2021) [37] | Amhara | 2021 | 352 | 92 | 7 | 22.2(17.86, 26.54) | Self-administered | Not reported |
| Bizuwork K et al, 2019 [36] | Amhara | 2019 | 143 | 100 | 6 | 32.9(25.20, 40.60) | Self-administered | Not Funded |
| Bikamo .E, 2021 [40] | Addis Ababa | | 215 | 96.5 | 7 | 41.5(34.91, 48.09) | Interview | Not reported |
| Benwu.MK, 2021[42] | Tigray | 2021 | 245 | 100 | 6 | 57.55(51.36, 63.74) | Interview | Not Funded |
| Abebaw M et al. . . 2022 [41] | Addis Ababa | 2022 | 409 | 100 | 7 | 87.3(84.07, 90.53) | Self-administered | Not funded |
| Bekele et al, 2021 [38] | Amhara | 2021 | 360 | 100 | 7 | 46.7(41.55, 51.85) | Self-administered | UOG |
| Sintayehu et al,2020 [43] | Harar | 2020 | 437 | 97.7 | 7 | 11.2(8.24, 14.16) | Oberservation | Not reported |
| Getuta, 2022 [17] | SNNP | 2020 | 422 | 100 | 7 | 31.8(27.36, 36.24) | Interview | Not reported |

f = 8); P < .001) indicating high heterogeneity. Therefore, random effects model was used to estimate the pooled prevalence of health professionals' knowledge on resuscitation in Ethiopia. The overall pooled knowledge prevalence was 47.6 (95% CI: 29.899, 65.300) (**Table 4** and **Fig 2**).

## Investigation of heterogeneity

Given the statistical heterogeneity of prevalence of knowledge of resuscitation outcome between the included primary studies ($I^2$ statistics = 99.21%), we performed subgroup analyses based on the following criteria: (a) the subgroup analysis hypothesis was pre-specified (a

**Table 4. Pooled prevalence of Health Professionals' knowledge on basic life support/resuscitation in Ethiopia, 2023.**

| Study | Effect size | 95% CI | % weight |
|---|---|---|---|
| Kelkay et al. 2018 | 38.60 | 33.811, 43.389 | 11.13 |
| Mersha A et al..,2020 | 76.20 | 72.243, 80.157 | 11.16 |
| Mersha AT, et al.., 2020 | 25.10 | 20.973, 29.227 | 11.15 |
| Abebe et al..2021 | 22.20 | 17.858, 26.542 | 11.15 |
| Bizuwork K et al, 2019 | 32.90 | 25.199, 40.601 | 10.99 |
| Bikamo .E, 2021 | 41.50 | 34.914, 48.086 | 11.05 |
| Benwu.MK, 2021 | 57.55 | 51.361, 63.739 | 11.07 |
| Abebaw M et al. . . 2022 | 87.30 | 84.073, 90.527 | 11.18 |
| Bekele et al, 2021 | 46.70 | 41.546, 51.854 | 11.12 |
| Theta(Overall) | 47.60 | 29.899, 65.300 | |

Heterogeneity: tau2 = 726.7778, I2 (%) = 99.21, $H^2$ = 126.90

Test of theta = 0: z = 5.27, Prob > |z| = 0.0000

Test of homogeneity: chi-square: 1015.18 with (d.f: 8), Prob > Q = 0.0000

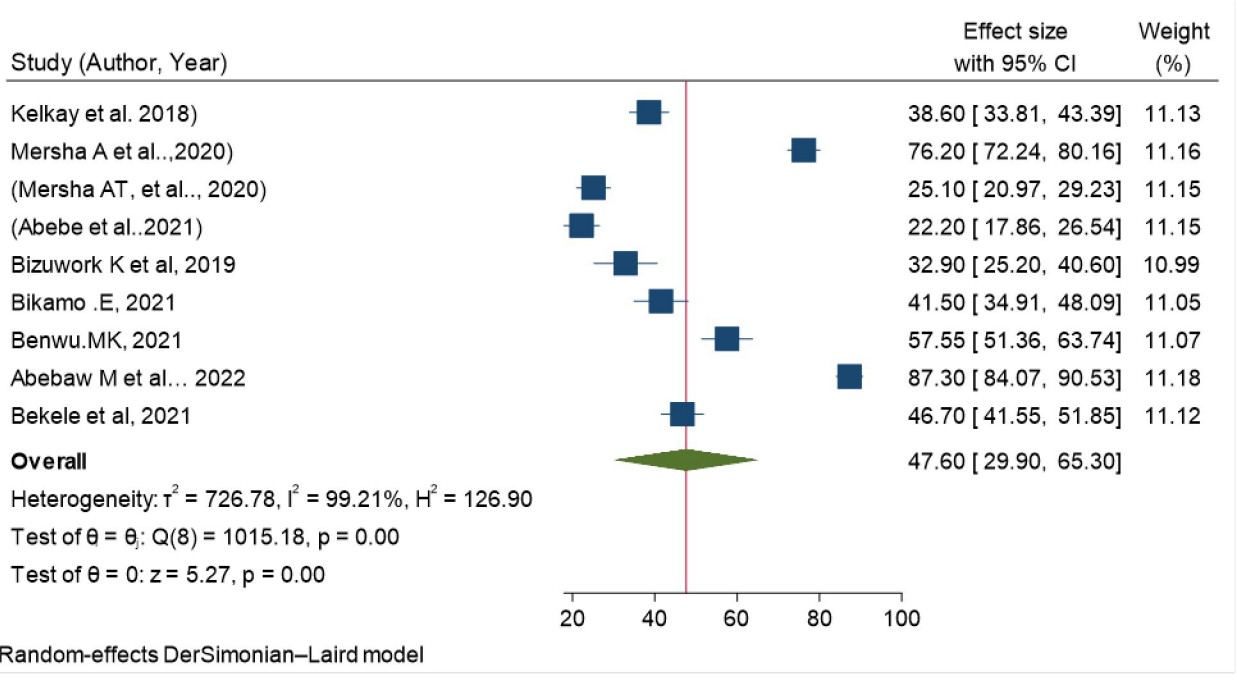

**Fig 2. The pooled prevalence of Health Professional's knowledge on basic life support / resuscitation in Ethiopia, 2023.**

priori) in the review protocol under consideration in PROSPERO; (b) there were large subgroup effect sizes; (c) Consistent interaction across the effect sizes (Prevalence) of knowledge outcome; and (d) the sub-grouping factors (region of study, and sample size were characteristics of interest measured at baseline across the studies. All the aforementioned criteria enabled us to place high confidence on the results of our subgroup analyses. Additionally, to explore the source of heterogeneity meta-funnel plot and egger stastical test were computed. The egger test revealed the absence of publication bias between the primary studies (P = 1.00). But, funnel plot also showed that there is slightly non-symmetrical distribution among primary studies (**Fig 3**).

## Subgroup analysis

The subgroup analysis based on region: From this analysis the highest pooled estimates of prevalence of Health Professional's knowledge on resuscitation was seen in SSNP 76.2(95% CI: 72.24, 80.16). Whereas the lowest pooled estimate was seen in Amhara region 33.04 (95% CI: 223.80, 42.27) (**Fig 4**).

## Sensitivity analysis for knowledge prevalence

Sensitivity analysis was done to identify the potential influence of single study on the pooled estimate of knowledge outcome. Using the random effects model, the result of sensitivity analysis suggested that the omission of 3 studies (Mersha AT et al, Abebe et al, and Abebaw et al) influenced the pooled estimate significantly. From the sensitivity result omission of Mersha AT et al increased the pooled estimate to 50.43 (32.15, 68.71) and the omission of Abebe et al also increased the pooled estimate to 50.79(32.85, 68.73). While, the omission of Abebaw et al was decreased the pooled estimate of knowledge outcome to 42.61(28.09, 57.13) (**Table 5** and **Fig 5**).

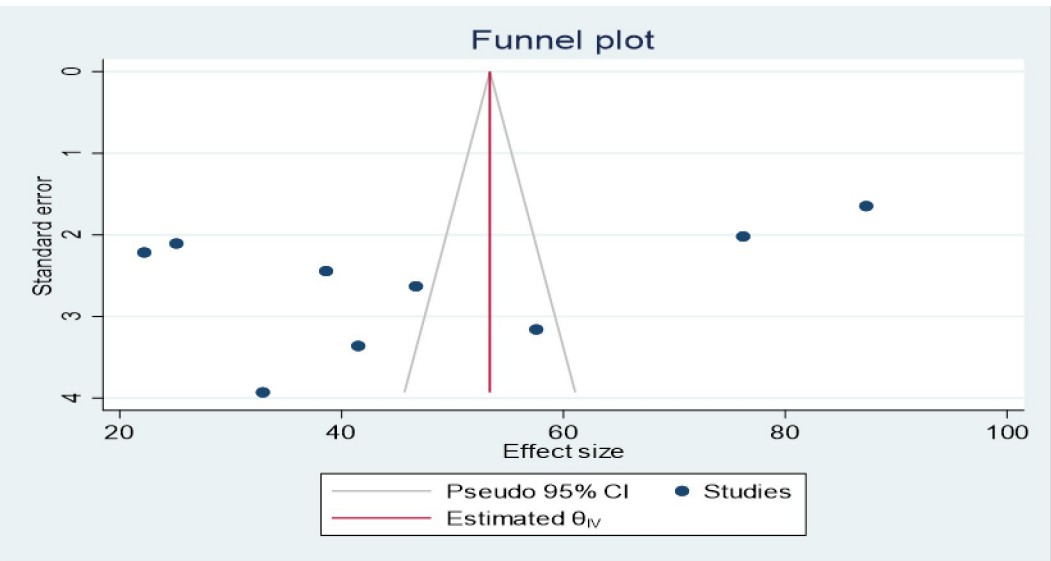

**Fig 3. The funnel plot of among 9 studies included in this systematic review and meta-analysis.**

## The pooled Health Professionals' practice on basic life support/resuscitation in Ethiopia

In this meta-analysis, the random effects model was used to estimate the pooled prevalence of resuscitation practice among Health Professionals. The overall pooled estimate of resuscitation practice among Health Professional's in Ethiopia was 44.42 (95% CI: 16.42, 72.41). Nevertheless, a considerable Heterogeneity among included studies was seen and objectively detected by $I^2$ stastics ($I^2$ = 99.69, P-value = 0.002) (**Table 6 and Fig 6**). Additionally, to explore the source of heterogeneity meta-funnel plot and egger stastical test were computed. The egger test revealed the absence of publication bias between the primary studies (P = 0.466). But, funnel plot also showed that there is slightly non-symmetrical distribution among primary studies (**Fig 7**).

## Sensitivity analysis

Sensitivity analysis was done to identify the potential influence of single study on the pooled estimate of Practice outcome. Using the random effects model, the result of sensitivity analysis suggested that the omission of 3 studies (Sintayehu et al. 2020, Benwu, KM. 2021, Abebaw M, et. al.2022) influenced the pooled estimate significantly. From the sensitivity result omission of Sintayehu et al. 2020 increased the pooled estimate practice to 49.98 (CI: 23.26, 76.70). While, the omission of Benwu, KM. 2021 and Abebaw M, et. al. 2022 was decreased the pooled estimate of Practice outcome to 36.84(7.79, 65.89) and 36.93(10.58, 63.29) (**Table 7 and Fig 8**).

## Factors associated with Health Professional's knowledge on basic life support/resuscitation in Ethiopia

Out of the included studies two studies (Kelkay et al..,2020, Mersha AT, et al.., 2020) explained that Health Professionals who had 5–10 yeas and above were knowledgeable regarding to basic life support/resuscitation. But, the pooled meta-analysis result, AOR: 1.781(0.846, 2.716),

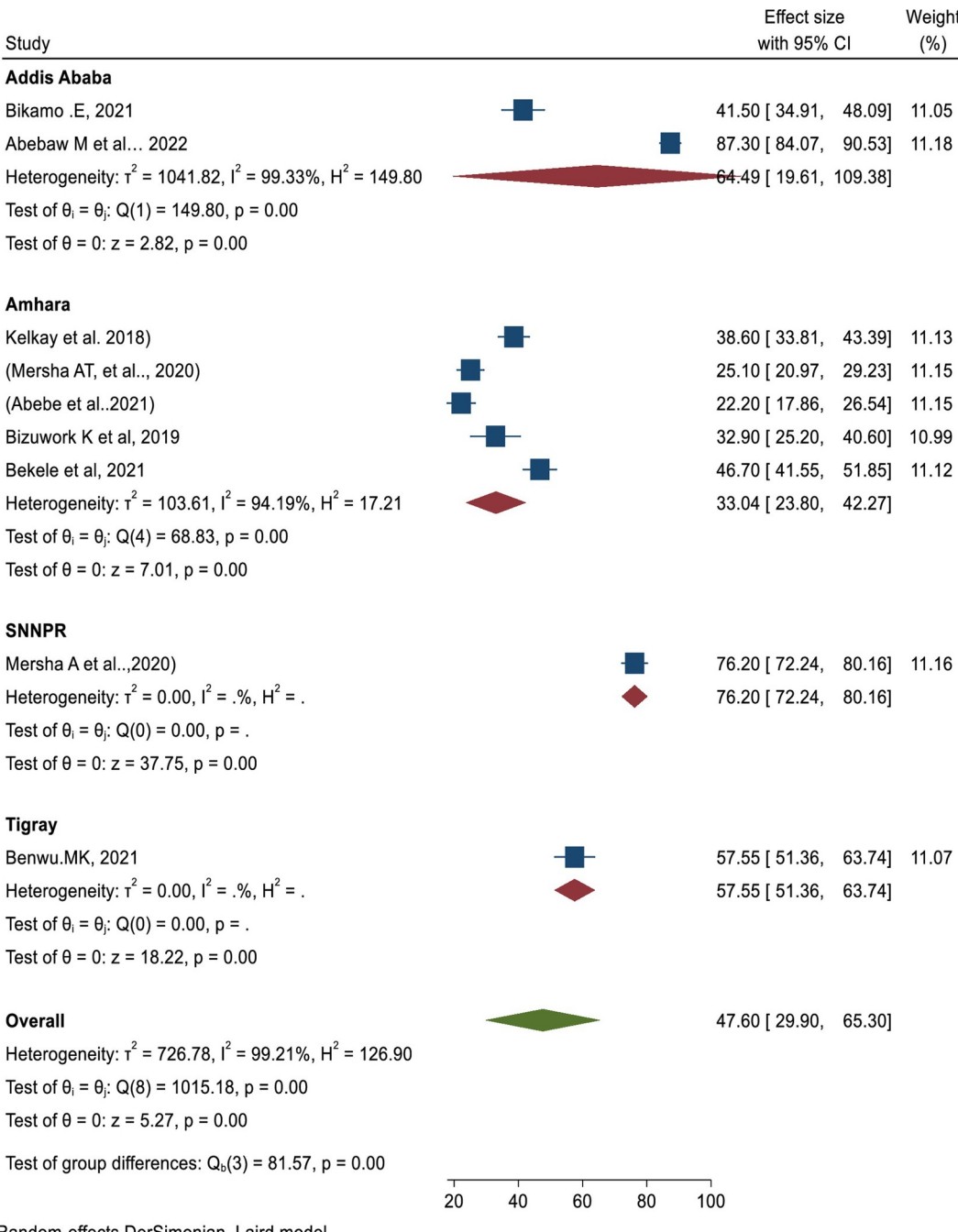

**Fig 4. The regional subgroup analysis of Health Professionals knowledge on basic life support /resuscitation in Ethiopia, 2023.**

showed the variable clinical experience has no an association with the knowledge of Health care providers regarding to resuscitation. However, the pooled meta-analysis result, AOR: 1.90 (1.24, 2.56) revealed that the educational status the professionals have a significant association with their knowledge of resuscitation (**Fig 9**).

Another four primary studies explained that the knowledge of health professional were associated with recently involving in resuscitation practice. But, the pooled meta-analysis

**Table 5. The sensitivity analysis of the pooled prevalence Health Professional's knowledge on basic life support/resuscitation, 2023, Ethiopia.**

| Omitted study | Effect size | 95% CI | P-value |
|---|---|---|---|
| Kelkay et al. 2018 | 48.723 | 29.052, 68.393 | 0.000 |
| Mersha A et al..,2020 | 44.006 | 24.943, 63.069 | 0.000 |
| Mersha AT, et al.., 2020 | 50.427 | 32.149, 68.705 | 0.000 |
| Abebe et al..2021 | 50.791 | 32.850, 68.733 | 0.000 |
| Bizuwork K et al, 2019 | 49.414 | 30.414, 68.414 | 0.000 |
| Bikamo .E, 2021 | 48.355 | 29.021, 67.689 | 0.000 |
| Benwu.MK, 2021 | 46.357 | 26.822, 65.893 | 0.000 |
| Abebaw M et al. . . 2022 | 42.607 | 28.089, 57.126 | 0.000 |
| Bekele et al, 2021 | 47.708 | 27.876, 67.539 | 0.000 |
| Combined | 47.600 | 29.899, 65.300 | 0.000 |

result revealed that recently involving in the resuscitation activity has no effects on the professional's knowledge on basic life support/resuscitation. Since, the confidence intervals of the odds ratio AOR (1.43 (0.70, 2.15)) include one the recently involving in the resuscitation activity is not significantly associated with the knowledge of Health professionals.

Additionally, six primary studies were revealed the training regarding to the basic life support was significantly associated with Health Professionals knowledge of resuscitation. But, pooled analysis AOR: 0.92 (0.11, 1.72) showed that training has no association with Professionals knowledge of basic life support.

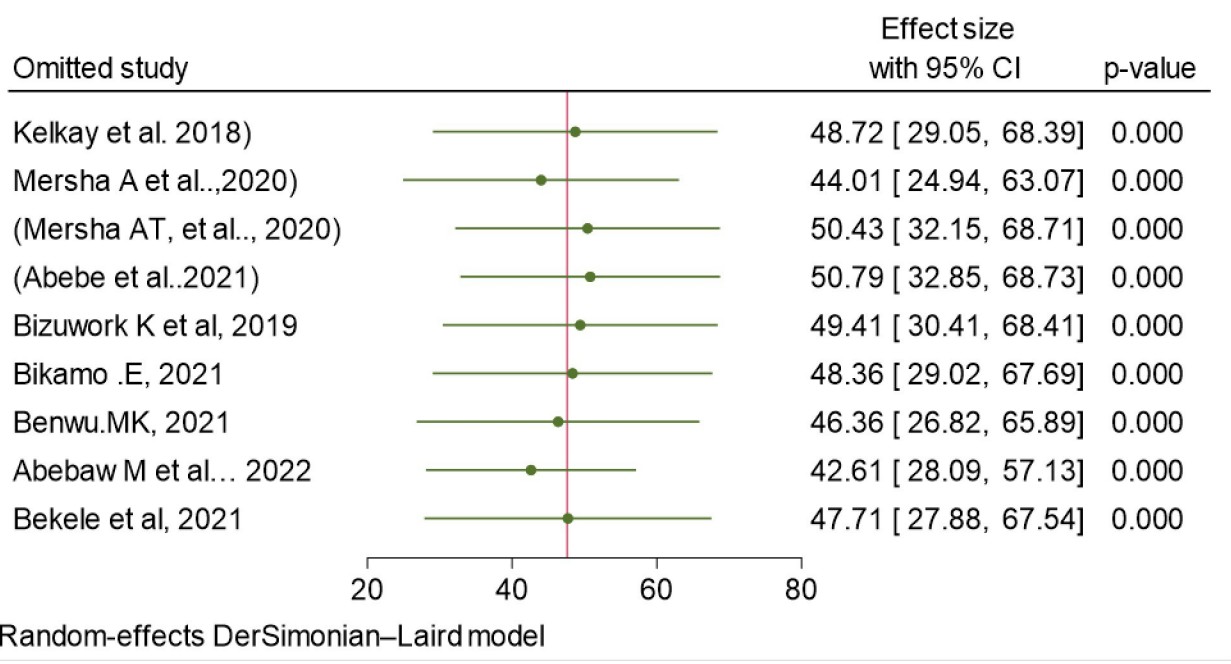

**Fig 5. Sensitivity analysis of the pooled prevalence of Health Professionals' knowledge on basic life support/resuscitation in Ethiopia, 2023.**

**Table 6. The pooled estimate of Health Professionals' practice on basic life support/ resuscitation in Ethiopia, 20203.**

| Study | Effect size | (95% CI) | % weight |
|---|---|---|---|
| Kelkay et al. 2018 | 28.40 | 23.96, 32.84 | 14.29 |
| Getuta, 2022 | 31.80 | 27.36, 36.24 | 14.29 |
| Bizuwork K . . .et al, 2019 | 24.50 | 17.45, 31.55 | 14.22 |
| Sintayehu et al,2020 | 11.20 | 8.24, 14.16 | 14.32 |
| Bikamo.E, 2021 | 35.80 | 29.39, 42.21 | 14.24 |
| Benwu,KM, 2021 | 89.80 | 86.01, 93.59 | 14.31 |
| Abebaw M et al . . .2022 | 89.20 | 86.19, 92.21 | 14.32 |
| Overall | 44.42 | 16.42, 72.41 | |

## Factors associated with the Health Professionals practice of basic life support/resuscitation in Ethiopia, 2023

The random effects model was used to identify factors that affect the pooled estimated practice of Health Professional's in Ethiopia. Some of the primary studies explained that educational status, recently involving in resuscitation activities and taking resuscitation training had significant association with practice outcome. But, the pooled estimated analysis revealed that Knowledge of participants AOR (0.99(0.48, 1.50)), taking resuscitation training AOR (1.06 (0.65, 1.46)), experience AOR (0.91(0.4, 1.42)), educational status AOR (1.23(0.34, 4.49)) were not significantly associated with practice outcome.

## Discussion

The pooled prevalence of Health Professional's knowledge about basic life support in Ethiopia was 47.6 (95% CI: 29.90, 65.30). This finding is similar with study conducted at Pakistan (41.7%) [44], Yemen (53.6%) [45], Nigeria 43.6% [46] and Botswana 48% [47]. While this finding is lower than study conducted at India 64% [48], Islamic Republic of Iran 69% [49]. This

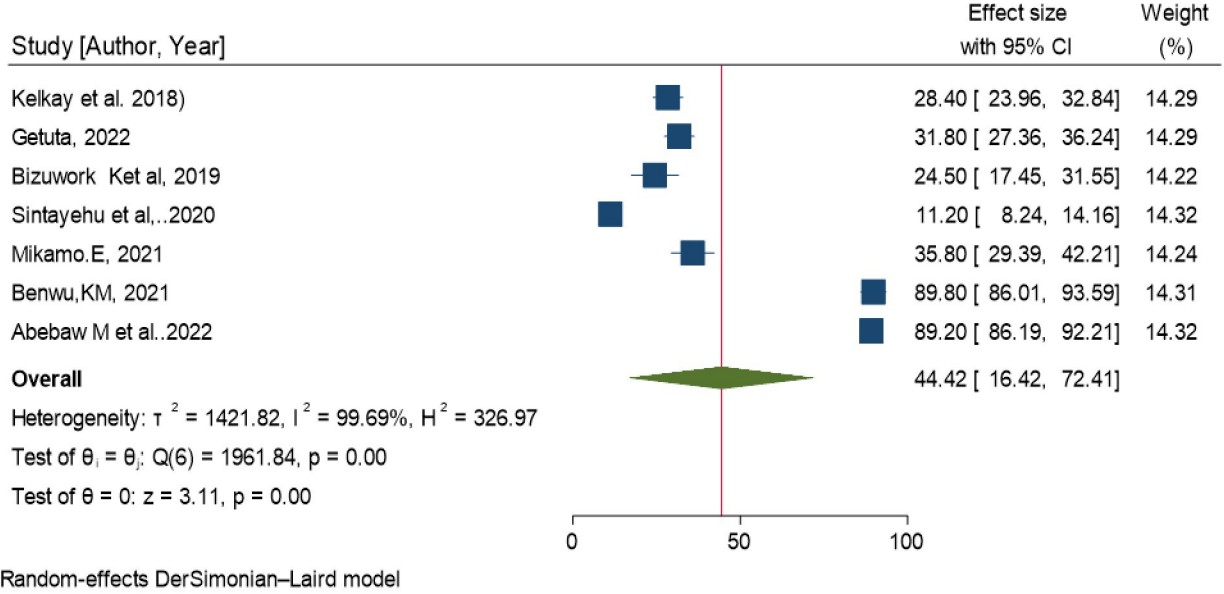

**Fig 6. The pooled Health Professional's practice on basic life support/resuscitation in Ethiopia, 2023.**

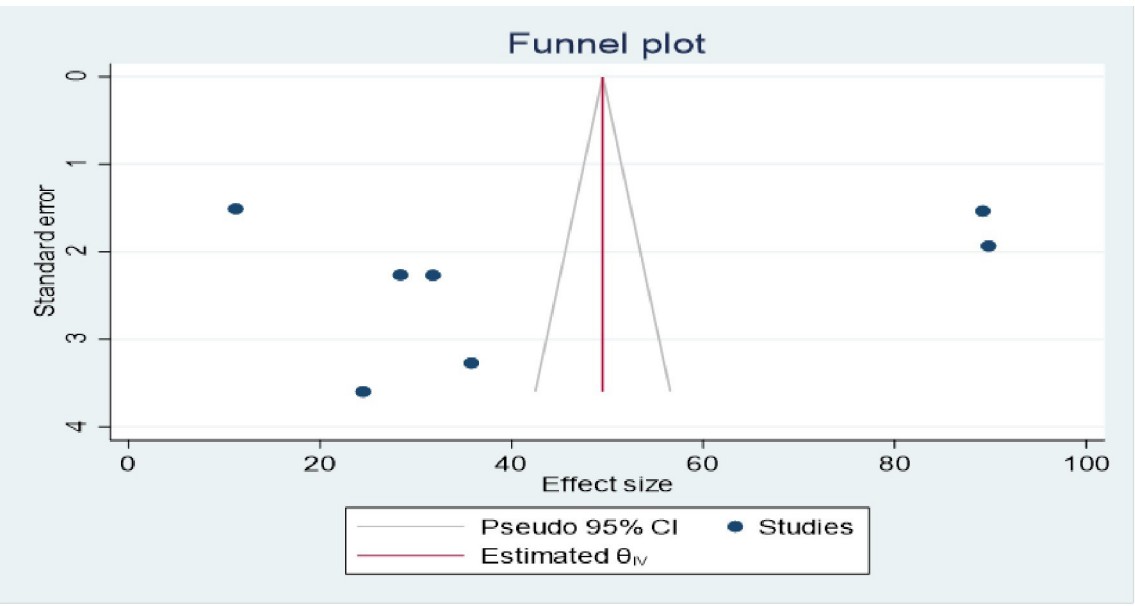

**Fig 7. The funnel plot for Health Professional's practice on basic life support/resuscitation in Ethiopia, 2023.**

discrepancy might due to stastical analysis (ANOVA, chi-square test), and geographical difference. Additionally, our finding was higher than study conducted at Egypt 33.92% [50].

In this study, pooled prevalence of Health professional's practice on Basic life support in Ethiopia was 44.42 (16.42, 72.41), this finding is in line with a study conducted at Uganda 46% [51]. But, this finding was lower than a study conducted at India 66% [48] and Nigeria 65.2% [52]. The difference might due to methodological difference, d study setting.

According to the final pooled analysis, educational status of the participant's was significantly associated with their knowledge of basic life support. Those professionals with degree and above was 1.9 time (AOR: 1.90(1.24, 2.56)) more likely knowledgeable regarding basic life support than under degree in educational status. This might be as educational level increase the Health professional's exposure to different procedure also increased.

Generally, this meta-analysis finding revealed that the Health professional's knowledge and practice on basic life support was poor. So, the concerned bodies should focus to enhance the

**Table 7. Sensitivity analysis of the pooled seven primary studies about prevalence of Health Professional's practice regarding basic life support in Ethiopia.**

| Omitted study | Effect size | (95% CI) | P-Value |
|---|---|---|---|
| Kelkay et al. 2018 | 47.08 | 15.06, 79.11 | 0.004 |
| Getuta, 2022 | 46.52 | 14.25, 78.78 | 0.005 |
| Bizuwork K et al, 2019 | 47.72 | 16.87, 78.56 | 0.002 |
| Sintayehu et al,2020 | 49.98 | 23.26, 76.70 | 0.000 |
| Bikamo.E, 2021 | 45.84 | 14.52, 77.17 | 0.004 |
| Benwu,KM, 2021 | 36.84 | 7.79, 65.89 | 0.013 |
| Abebaw M et al.2022 | 36.93 | 10.58, 63.29 | 0.006 |
| Combined | 44.42 | 16.42, 72.41 | 0.002 |

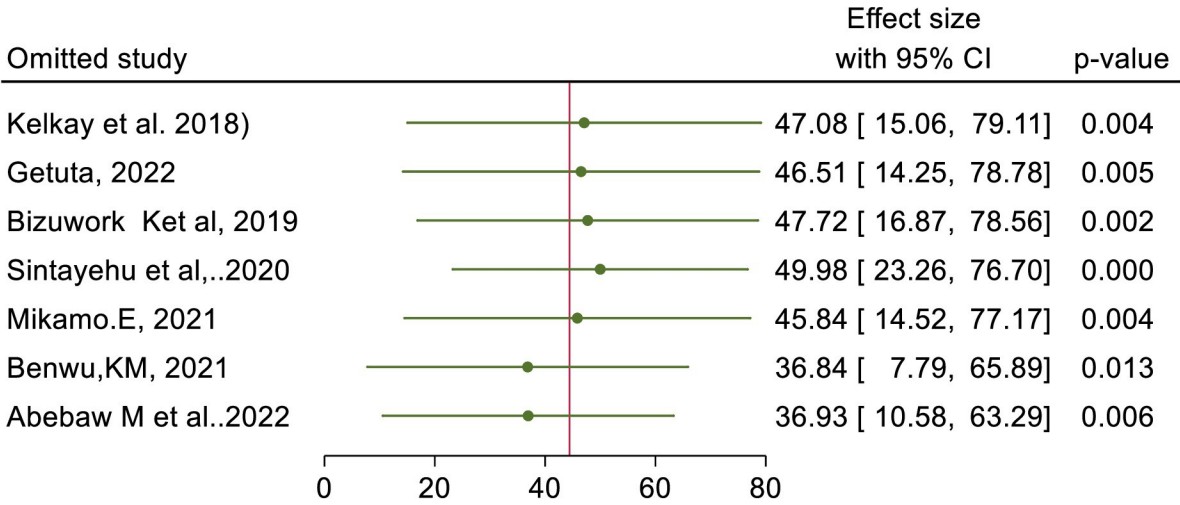

**Fig 8. The sensitivity of the pooled prevalence of Health Professional's practice regarding basic life support/resuscitation in Ethiopia, 2023.**

professional's capacity regarding on the implementation of basic life support at Health facilities in Ethiopia.

## Limitation

This systematic review and meta-analysis had highly considerable heterogeneity this might affect pooled estimate of the outcome. Similarly, there was small size in the included studies this also had effects on the pooled generalization. Additionally, all the included studies were also used cross sectional study design which might affects the representativeness of the pooled estimates for the target populations.

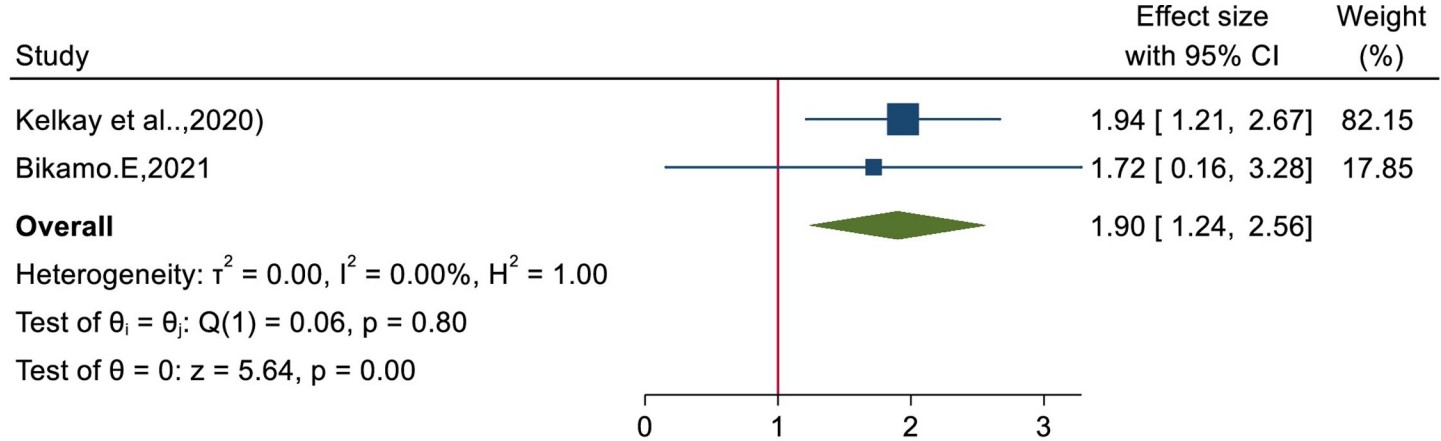

Random-effects DerSimonian–Laird model

**Fig 9. Forest plot for the association of educational status and professional's knowledge on basic life support in Ethiopia, 2023.**

### Strength of the review

This systematic review is conducted for the first time in Ethiopia. This systematic review includes team members from different disciplines and conduct sensitivity analysis, subgroup analysis to dissolve heterogeneity among included studies. Additionally, this review also tried to incorporate all studies regarding to the research question. The risk assessments and quality assessment for each study also conducted.

## Conclusion

In conclusion, the overall pooled prevalence of Health professional's knowledge and practice on basic life support in Ethiopia was 47.6% and 44.42% respectively. This finding was very poor to save the life of critically ill patients at emergency site. Only the educational status of the professional's was significantly associated with knowledge outcomes. Moreover, Health Professionals and responsible stakeholders should give focus and advance on the knowledge and practice of basic life support/CPR in Ethiopia.

## Supporting information

**S1 Checklist. PRISMA 2020 Checklist.**
(DOCX)

**S1 Data.**
(XLSX)

**S2 Data.**
(XLSX)

## Author Contributions

**Conceptualization:** Worku Necho Asferie, Demewoz Kefale, Amare Kassaw, Amare Simegn Ayele, Gedefaye Nibret, Habitamu Shimels Hailemeskel, Solomon Demis, Tigabu Munye Aytenew.

**Data curation:** Worku Necho Asferie, Demewoz Kefale, Gedefaye Nibret, Yohannes Tesfahun, Habitamu Shimels Hailemeskel, Solomon Demis, Shegaw Zeleke, Tigabu Munye Aytenew.

**Formal analysis:** Worku Necho Asferie, Amare Kassaw, Solomon Demis.

**Methodology:** Worku Necho Asferie, Amare Simegn Ayele, Shegaw Zeleke.

**Project administration:** Demewoz Kefale.

**Software:** Amare Kassaw, Shegaw Zeleke.

**Supervision:** Amare Simegn Ayele, Habitamu Shimels Hailemeskel, Tigabu Munye Aytenew.

**Validation:** Tigabu Munye Aytenew.

**Writing – original draft:** Worku Necho Asferie, Yohannes Tesfahun, Tigabu Munye Aytenew.

**Writing – review & editing:** Worku Necho Asferie, Demewoz Kefale, Gedefaye Nibret, Yohannes Tesfahun, Habitamu Shimels Hailemeskel, Tigabu Munye Aytenew.

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
