## [Decision Letter · Decision Letter 0]

19 Sep 2023

PONE-D-23-16248Health Professionals’ knowledge and Practice on Basic Life Support and its predicting factors in Ethiopia: Systematic Review and Meta-analysisPLOS ONE

Dear Dr. Asfere,

Thank you for submitting your manuscript to PLOS ONE. After careful consideration, we feel that it has merit but does not fully meet PLOS ONE’s publication criteria as it currently stands. Therefore, we invite you to submit a revised version of the manuscript that addresses the points raised during the review process.

We look forward to receiving your revised manuscript.

Kind regards,

Ayele Mamo Abebe, MSc in pediatric and child health nursing

Academic Editor

PLOS ONE

Journal Requirements:

3. Please upload a copy of Figures 1, 2 and 4 to which you refer in your text on page xx. If the figure is no longer to be included as part of the submission please remove all reference to it within the text.

Additional Editor Comments :

It needs major revision.

Reviewers' comments:

Reviewer's Responses to Questions

**Comments to the Author**

1. Is the manuscript technically sound, and do the data support the conclusions?

Reviewer #1: Yes

Reviewer #2: Partly

Reviewer #3: Partly

2. Has the statistical analysis been performed appropriately and rigorously? 

Reviewer #1: Yes

Reviewer #2: No

Reviewer #3: No

3. Have the authors made all data underlying the findings in their manuscript fully available?

Reviewer #1: Yes

Reviewer #2: No

Reviewer #3: Yes

4. Is the manuscript presented in an intelligible fashion and written in standard English?

Reviewer #1: Yes

Reviewer #2: No

Reviewer #3: No

5. Review Comments to the Author

Reviewer #1: Congratulations authors on your scholarly work; you have brought an important study problem with good findings that have public health importance in the area of practice. However, there are some minor issues that I want you to address before considering the manuscript for publication.

General comment

There are several typological and grammar usage errors that need extensive proof reading for revisions.

Specific comments

Methods

I suggest the authors consider using adapted PICO (possibly with the T or S) format to make the review question explicit and assist with clear specification of the inclusion and exclusion criteria.

Provide the primary search string, including the truncation and synonyms as a supplementary file for at least one database. Specify the type of searching strategy (e.g. was it line by line, by combined concepts, did the search include title (TI), abstract (Ab) or full text or all these categories) was used, specify if databases were searched independently and if any modifications were made to the search strategy (e.g. limiters) for different databases. The description of the method should include enough details for reproducibility. Explain the steps in screening (e.g. title, abstract, full text).

I suggest reporting the statistics for measurement of the level of agreement for the independent reviews (e.g. quality scores) of each article. Moreover, kindly append a table showing methodological quality of the appraised articles with the last column being ‘overall quality score’.

How did you pool the individual knowledge, and practice scores because nearly all the primary studies used to consider their own operational definitions? Please be critical in this issue and support your measurement of these pooled outcomes with sound references.

Explain the data extracted from the studies, reliability and validity of data extraction tool. Explain how the variables extracted were determined and criteria for data to be suitable for extraction. Explain if data transformation was required or undertaken when data were reported differently.

Results

Please, use PRISMA 2020 flow diagram. https://prisma-statement.org/PRISMAStatement/FlowDiagram

Please include two columns in Table 1: data collection technique (interview, observation, self administered questionnaire, etc) and funding source for each study (You can say not funded, not reported or name of funder if funded).

Reviewer #2: It is a great pleasure for me to get the opportunity to review a manuscript entitled ‘Health Professionals’ knowledge and Practice on Basic Life Support and its predicting factors in Ethiopia: Systematic Review and Meta-analysis’

I would like to appreciate the authors for their efforts go this far to investigate the impacts of professionals’ knowledge, practice and associated factors on Basic life support. However, I have some concerns that need to be address before providing my recommendation for publication.

Title

The tittle seems fine, but I personally don’t prefer the factor to be on the title as primary study rather you better include meta-regression for covariates and factor analysis could be described in data analysis section along with sub-group analysis and meta-regression.

Abstract:

Background section: it could incorporate the summary of the magnitude of BLS, discrepancies globally in global context, and justification of your review in 3-4 lines.

Method: A comprehensive search strategy was conducted in…….. From ……to….with/without language restrictions or without date and language restriction; No database called international or you may use big databases if you like to say so.

Conclusion: ‘how low is considerably low?’

Introduction

I am sorry to say but this section is poorly written part of the manuscript. It lacks components that needs to be addressed in this section which might include but not limited to description of the problem, risk factors, etiology, impacts, outcomes

There are plenty of questions that need to be addressed:

- What are the magnitude of cardiopulmonary arrest, and other problems that could be addressed with basic life support from global to local context?

- What are gaps you identified during your literature search?

- Why did you pan this review?

- What is the prevalence of low knowledge level of BLS among health professionals from the available literatures globally?

- What is the prevalence of low practical skill of BLS among health professionals from the available literatures globally?

- With whom these skills are very low the most?

Method

The method section subheadings should be organized as Protocol and registration, Eligibility criteria, Comprehensive search, Selection of studies, Critical appraisal, Data extraction…….

The search should be comprehensive and you have to report a complete comprehensive search strategy at least for one database. Where and how did you get the Mesh terms for PubMed and other databases?? How knowledge and skill was measured in the primary studies?? How valid the cut point was?

Data-analysis

Subgroup analysis could be conducted by categorical moderators such as health care provider, year of experience, study setting etc. and meta-regression on covariates such as mean age, experience, sample size, quality score, and others. Besides, factor analysis better be conducted by age, profession, experience, gender, and other moderators

Result

The result lacks comprehensiveness where it could include the summary of search result, and description of the included studies in terms of mean age, sample size, profession, study setting, and how these studies were addressed specifically for practice assessment which might have biases.

Meta-analysis

The met-analysis better could be done with metaprop command rather than drop down menu as it has a wide margin of safety. The factor analysis effect estimate should be reported with AOR with 95% CI on the forest plot

Discussion

The discussion is very shallow with limited comparison and impact analysis of your finding with the available literatures.

The discussion should be finalized with the following subheadings

- Limitation and strength

- Implication for policy-makers

- Implication for clinical practice

- Implication for further studies

Generally, I would say the topic is very relevant, but the authors have a lot to do.

Reviewer #3: Cardiac arrest is a common cardiac problem leading to sudden death of patients. Basic life support is a first aid management that is important for managing cardiac arrest. The knowledge and skill of health care professionals have paramount importance for effectively practicing Basic life support. Though you have presented us an important figure you haven’t extracted important variables especially in the associated factor section. To make your articles stronger try to include the most important variables from the articles. The discussion section needs more work to make your paper outstanding. Please have some of my comments below. Thanks!

In the background section

• Clearly show the extent of the problems. The magnitude of patient suffering from cardiac arrest in Africa and in Ethiopia.

• Also, include how much CPR is effective in managing cardiac arrest compared to other methods

Method Section

• The two investigators (WN and TM) independently and carefully reviewed the contents of each retrieved articles. Those literatures fulfilling the following criteria were finally included in the review.....Is this paragraph describe who searched the article or who evaluated the literature for eligibility criteria....either move to eligibility criteria sub section or remove it from here

• Be consistent when you describing the authors....WN verses W.N.W

• Please operationally define the outcome variable. How knowledge and practice were measured in the included articles. The tools used for measuring the outcome variable. Is there any difference in the tools used by individual studies included?

• Please include the reference for JBI appraisal tool.

• in the data extraction you haven't mentioned how you extracted the variables for associated factors.

• please include the references for the cut point for I2...from where you have taken the cut point for deciding the heterogeneity.

Result Section

• Addis Ababa...but not Addis Abeba

• Table 2 ….the method section lacks how you evaluated the quality assessment. The description of the tool used for quality assessment. Please include in method section.

• In the knowledge you used only 9 articles out of 11. Why you excluded 2 articles? If you have a reason this should be described in the study characteristics section.

• Are the two terms Basic life Support/Resuscitation were synonyms? do you think the articles measured this two terms in similar way?? This needs clarification in the method section.

• I am not clear with the statement you written here...The presence of heterogeneity was already identified by I2. To identify the source of heterogeneity we perform meta regression and subgroup analysis. Here you conducted subgroup analysis based on the regions in Ethiopia. So, what is you rational behind listing the criteria to list here. please include the literature for your elaboration. I need citation for this section.

• Publication bias/small study effects these two terms were different in the way they describe bias. Also, they have different statistical method for calculating Publication bias and small study effects? please describe in the method section how they were determined in this article.

• The pooled Health professionals’ Practice on basic life support you only included seven articles out of 11. Why?? needs description. Which articles were excluded from computing practice.

• There is no text description for fig 7. Also, you haven't conducted the Publication bias and small study effects for Health professionals’ Practice on basic life support??

• For factors associated ………Please include a table describing which the variables you extracted from individual studies.

• Which category of educational status were knowledgeable on resuscitation ???

Discussion Section

• remove the first paragraph from discussion section or either include in the background section.

• It is not possible to discuss meta-analysis with single study. rather find similar meta-analysis for comparing your finding.

• write the strength and limitation in the final paragraph of discussion section.

6. PLOS authors have the option to publish the peer review history of their article (what does this mean?). If published, this will include your full peer review and any attached files.

Reviewer #1: No

Reviewer #2: No

Reviewer #3: No

---

## [Author Response · Author response to Decision Letter 0]

6 Nov 2023

Response for the given Comments

 For Academic Editor

1. Please ensure that your manuscript meets PLOS ONE's style requirements? 

 The manuscript was corrected with considering the Plos One journal manuscript preparation templates

2. In your Data Availability statement, you have not specified where the minimal data set underlying the results described in your manuscript can be found.

 The minimal data set was uploaded as supporting information 

3. Please upload a copy of Figures 1, 2 and 4 to which you refer in your text on page xx. If the figure is no longer to be included as parts of the submission please remove all reference to it within the text.

 The requested figures were upload in “list of figures” separately from the manuscript

2. Reviewer #1

 General comment: There are several typological and grammar usage errors that need extensive proof reading for revisions.

 We tried to consider for correcting the grammar and typographic errors, the corrected one was highlighted on the revised manuscript.

 Specific Comments

Method

1. I suggest the authors consider using adapted PICO (possibly with the T or S) format to make the review question explicit and assist with clear specification of the inclusion and exclusion criteria.

 We the authors consider the utilization of adapted PICO from previously published systematic articles.

2. Provide the primary search string, including the truncation and synonyms as a supplementary file for at least one database. Specify the type of searching strategy (e.g. was it line by line, by combined concepts, did the search include title (TI), abstract (Ab) or full text or all these categories) was used, specify if databases were searched independently and if any modifications were made to the search strategy (e.g. limiters) for different databases. The description of the method should include enough details for reproducibility. Explain the steps in screening (e.g. title, abstract, full text).

 We search the relevant articles based on inclusion criteria by using the following steps like line by line, combining the MeSH Terms using Booleans terms, filters and truncation. The MeSH terms and limiters were used to retrieve the relevant articles. Then, the reference the articles of the article which fit the interest of the review were further assessed and selected if they full fill the inclusion criteria. Some of the Boolean used were “AND”, “OR”, and limiters are also year, human, observational studies. 

 First the title was seen if full filling the abstract was thoroughly screened and then the those abstracts which was including the reviewed questions, stastical analysis and methods of estimation fitted with reviewing protocol it was selected for further full artic review. Further the search details will be uploaded as supplementary file as requested. 

3. How did you pool the individual knowledge, and practice scores because nearly all the primary studies used to consider their own operational definitions? Please be critical in this issue and support your measurement of these pooled outcomes with sound references.

 The individual primary articles had their own cut off point to categories level of knowledge and practice on the basic life support. We authors take the minimum cutoff point for both outcomes to pool all articles that include topics of this systematic review. Further see page 8.

4. Explain the data extracted from the studies, reliability and validity of data extraction tool. Explain how the variables extracted were determined and criteria for data to be suitable for extraction. Explain if data transformation was required or undertaken when data were reported differently.

 The data extraction form was adapted from Cochrane data extraction tool and used to extract the required data for our systematic review. The variables that full fill the inclusion criteria for data extraction was pooled for this study. The data transformation was not dome because almost all retrieved articles had similar data analysis methods to identify the outcome variable for our interested question. 

Result 

1. Please, use PRISMA 2020 flow diagram. 

 The PRISMA 2020 was already used to show flow diagrams

2. Please include two columns in Table 1: data collection technique (interview, observation, self-administered questionnaire, e.t.c) and funding source for each study (You can say not funded, not reported or name of funder if funded).

 The given comment was incorporated in Table 1, see further page: 11

Reviewer #2

Back Ground section

1. Clearly show the extent of the problems. The magnitude of patient suffering from cardiac arrest in Africa and in Ethiopia.

 The given comment was incorporated to the revised manuscript.

2. Also, include how much CPR is effective in managing cardiac arrest compared to other methods

Method Section

3. The two investigators (WN and TM) independently and carefully reviewed the contents of each retrieved articles. Those literatures fulfilling the following criteria were finally included in the review.....Is this paragraph describe who searched the article or who evaluated the literature for eligibility criteria....either move to eligibility criteria sub section or remove it from here.

 The given comment was corrected on the revised manuscript

4. Be consistent when you describing the authors....WN verses W.N.W

5. Please operationally define the outcome variable. How knowledge and practice were measured in the included articles. The tools used for measuring the outcome variable. Is there any difference in the tools used by individual studies included?

 We our incorporate the operational definition of the reviewed question in the method parts and individual studies used different cut of point to identify their outcome variable. We used the minimal cut off point among the retrieved articles to compute this systematic review. For further evidence see page 8 in the highlighted parts of the revised manuscript.

6. Please include the reference for JBI appraisal tool

 The JBI appraisal tool was cited on the revised manuscript

7. In the data extraction you haven't mentioned how you extracted the variables for associated factors.

 The given component was incorporated in the revised manuscript. See further page 9 in the highlighted manuscript.

8. Please include the references for the cut point for i2...from where you have taken the cut point for deciding the heterogeneity.

 The cut point to decide the Heterogeneity was taken from the Cochrane systematic review manual.

Result Section

1. Table 2 ….the method section lacks how you evaluated the quality assessment. The description of the tool used for quality assessment. Please include in method section.

 The criteria to appraise the quality each included articles already stated at Page 8 – 9 in the revised manuscript

2. In the knowledge you used only 9 articles out of 11. Why you excluded 2 articles? If you have a reason this should be described in the study characteristics section.

 The two articles were excluded in the knowledge review because they only stated the practice outcome. The articles that were excluded for knowledge outcomes were Sintayehu et al...2020 and Guteta 2022.

3. Is the two terms Basic life Support/Resuscitation were synonyms? Do you think the articles measured these two terms in similar way? This needs clarification in the method section.

 The two word most of the used exchangeable in concept of saving the life the patient who faced sudden respiratory and cardiac problem. Some studies used basic life support and others used basic resuscitation, they also follow slightly some difference in measurements but it has significance difference to interpret the outcome. The clarification was also incorporated the method section. See page 7

4. I am not clear with the statement you written here...The presence of heterogeneity were already identified by I2. To identify the source of heterogeneity we perform meta-regression and subgroup analysis. Here you conducted subgroup analysis based on the regions in Ethiopia. So, what are you rational behind listing the criteria to list here? Please include the literature for your elaboration. I need citation for this section.

 The systematic review and meta-analysis guide said before preceding to the review it is better to compute and look the Heterogeneity among the included studies (by looking the I2 and Q2). This evidence was supported by the Cochrane systematic review and meta-analysis manual. If there is any considerable heterogeneity, it is better to identify the source of heterogeneity. These findings help us to dissolve heterogeneity. This evidence was cited on the revised manuscript. See page 

5. Publication bias/small study effects these two terms were different in the way they describe bias. Also, they have different statistical method for calculating Publication bias and small study effects? Please describe in the method section how they were determined in this article.

 The way how to compute the publication bias and small study effect is stated at the manuscript see page 10.

6. The pooled Health professionals’ Practice on basic life support you only included seven articles out of 11. Why?? Needs description. Which articles were excluded from computing practice? 

 Among the included 11 studies only seven studies had practice outcomes; these articles were Mersha A et al..2020, Mersha AT et al, 2020, Bekele FA et al, 2021, and Abebaw M, 2022.

7. There is no text description for fig 7. Also, you haven't conducted the Publication bias and small study effects for Health professionals’ Practice on basic life support??

 The Publication bias and small study effect was computed for practice outcome. See page 20 

8. For factors associated ………Please include a table describing which the variables you extracted from individual studies.

 The table showed the sensitivity analysis of the pooled prevalence of practice outcome. See page 20.

9. Which category of educational status was knowledgeable on resuscitation???

 Those Health Professionals having BSc degree and above

2. Discussion Section

1. Remove the first paragraph from discussion section or either include in the background section.

 The given comments already corrected 

2. Write the strength and limitation in the final paragraph of discussion section.

 The strength and the limitation of the study was stated in the revised manuscript see page

---

## [Editor Report · Decision Letter 1]

5 Jan 2024

Health Professionals’ knowledge and Practice on Basic Life Support and its predicting factors in Ethiopia: Systematic Review and Meta-analysis

PONE-D-23-16248R1

Dear authors

We’re pleased to inform you that your manuscript has been judged scientifically suitable for publication and will be formally accepted for publication once it meets all outstanding technical requirements.

Kind regards,

Ayele Mamo Abebe, MSc in pediatric and child health nursing

Academic Editor

PLOS ONE

---

## [Editor Report · Acceptance letter]

22 Jan 2024

PONE-D-23-16248R1 

PLOS ONE

Dear Dr. Asferie, 

I'm pleased to inform you that your manuscript has been deemed suitable for publication in PLOS ONE. Congratulations! Your manuscript is now being handed over to our production team.

Kind regards, 

on behalf of

Assistant professor Ayele Mamo Abebe 

Academic Editor

PLOS ONE